# The Identification and Comparative Analysis of Non-Coding RNAs in Spores and Mycelia of *Penicillium expansum*

**DOI:** 10.3390/jof9100999

**Published:** 2023-10-09

**Authors:** Tongfei Lai, Qinru Yu, Jingjing Pan, Jingjing Wang, Zhenxing Tang, Xuelian Bai, Lue Shi, Ting Zhou

**Affiliations:** 1College of Life and Environmental Science, Hangzhou Normal University, Hangzhou 310036, China; laitongfei@hznu.edu.cn (T.L.); 2021111010050@stu.hznu.edu.cn (Q.Y.); 2022111010065@stu.hznu.edu.cn (J.P.); 2022111010055@stu.hznu.cn (J.W.); baixl2012@163.com (X.B.); shilue@126.com (L.S.); 2School of Culinary Arts, Tourism College of Zhejiang, Hangzhou 311231, China; tangzhenxing@126.com

**Keywords:** *Penicillium expansum*, development, transcriptome, non-coding RNA, miR168

## Abstract

*Penicillium expansum* is the most popular post-harvest pathogen and causes blue mold disease in pome fruit and leads to significant economic losses worldwide every year. However, the fundamental regulation mechanisms of growth in *P. expansum* are unclear. Recently, non-coding RNAs (ncRNAs) have attracted more attention due to critical roles in normalizing gene expression and maintaining cellular genotypes in organisms. However, the research related to ncRNAs in *P. expansum* have not been reported. Therefore, to provide an overview of ncRNAs on composition, distribution, expression changes, and potential targets in the growth process, a comparative transcriptomic analysis was performed on spores and mycelia of *P. expansum* in the present study. A total of 2595 novel mRNAs, 3362 long non-coding RNAs (lncRNAs), 10 novel microRNAs (miRNAs), 86 novel small interfering RNAs (siRNAs), and 11,238 circular RNAs (circRNAs) were predicted and quantified. Of these, 1482 novel mRNAs, 5987 known mRNAs, 2047 lncRNAs, 40 miRNAs, 38 novel siRNAs, and 9235 circRNAs were differentially expressed (DE) in response to the different development stages. Afterward, the involved functions and pathways of DE RNAs were revealed via Gene Ontology (GO) and Kyoto Encyclopedia of Genes and Genomes (KEGG) database enrichment analysis. The interaction networks between mRNAs, lncRNAs, and miRNAs were also predicted based on their correlation coefficient of expression profiles. Among them, it was found that miR168 family members may play important roles in fungal growth due to their central location in the network. These findings will contribute to a better understanding on regulation machinery at the RNA level on fungal growth and provide a theoretical basis to develop novel control strategies against *P. expansum*.

## 1. Introduction

*Penicillium expansum* is the major causal agent of pome fruit decay during post-harvest transportation and storage, which can lead to enormous economic losses each year around the world. Meanwhile, *P. expansum* can produce mycotoxins, such as patulin and citrinin, that influence fruit processing and raise the risk of a series of diseases in consumers [1,2]. *P. expansum* strains have been isolated from many regions and multiple hosts. Meanwhile, hazards induced by *P. expansum* have been taken more seriously [3,4]. The application of synthetic fungicides, such as imazalil, fludioxonil, thiabendazole, benomyl, and pyrimethanil, is the primary way to control *P. expansum*. However, fungicide contamination in the environment and the emergence of resistant strains are prompting scientists to find more effective and eco-friendly strategies [5]. The virulence of the pathogen is closely related to hosts (species and developmental stages) and environmental conditions (temperature, humidity, light type and intensity, atmosphere composition, pH, etc.) [6,7,8,9,10]. Physical control approaches generally aim to optimize environmental parameters to balance inhibitory efficiency and energy consumption. Acting as safe disinfectants, chemical compounds such as essential oils, phytohormones, chemical excitons, and generally recognized as safe (GRAS) substances have been explored and show extensive prosperity, as well as many antagonistic microorganisms. In addition, synergistic uses with physical or chemical methods are also being evaluated in many scenarios [5,11]. Nevertheless, to a certain extent, these new approaches have not met the commercial application requirement due to their own limitations.

To develop more excellent management strategies, the fundamental infection mechanisms of *P. expansum* have been studied for many years, and recent advances have been well reviewed by Wang et al. (2023) [12]. During the penetration process, *P. expansum* significantly change on the morphological and physiological level and produce various hydrolases and organic acids to damage the host cell wall and impair the host defense response [13,14]. Zhao et al. (2022) found that nine transcription factors were closely related to pears being infected with *P. expansum* [15]. A large number of genes, such as *PeRacA*, *PeRmtC*, *PeSCP*, *PeSntB*, *PeNIP*, *PeNox*, *PeSet1*, *PeLae*, *PeVeA*, *PeCreA*, *PeLysMs*, *PeNLPs*, and *PePRT*, were also proven to be critical to the virulence of *P. expansum* [16,17,18,19,20,21,22,23]. In addition, the biosynthesis pathway of patulin, which is the most important secondary metabolite of *P. expansum* and poses severe genotoxicity and cytotoxicity to animals and humans, has been systematically dissected at the molecular and biochemical level [1,24,25]. However, there are more in-depth and complex questions regarding the regulation of the virulence and vitality to be answered.

In recent years, non-coding RNAs (ncRNAs) have attracted more attention in a broad range of physiological and pathological processes, although they are not translated into proteins. As one of the components of epigenetic regulation, many remarkable works have demonstrated that ncRNAs play critical roles in normalizing gene expression and maintaining cellular genotypes [26]. For example, long non-coding RNAs (lncRNAs), which are longer than 200 nucleotides, can regulate gene expression due to individual domains that enable them to establish relationships with DNA, RNA, or protein [27,28]. MicroRNAs (miRNAs) and small interfering RNAs (siRNAs) can trigger RNA-induced interference via elaborate post-transcriptional mechanisms [29,30,31]. Circular RNAs (circRNAs), which are covalently closed single-stranded RNAs, have multiple functions, including modulating gene expression, binding proteins, adsorbing miRNAs, etc. [32]. Due to the high abundance in transcripts and emerging evidence of versatility in human beings, animals, plants, parasites, yeasts, and bacteria, ncRNAs are doubtless associated with the virulence and vitality of plant pathogenic fungi [33]. Avrova et al. (2007) found a non-protein-coding gene family called Pinci1 containing more than 400 members in *Phytophthora infestans*, which were significantly up-regulated during the biotrophic phase of infection [34]. Till et al. (2018) identified an lncRNA termed *HAX1* that was associated the cellulase expression in *Trichoderma reesei* QM9414 [35]. *Botrytis cinerea* could utilize the small RNAs Bc-siR3.1, Bc-siR3.2, and Bc-siR5 to suppress Arabidopsis-immunity genes by hijacking the AGO1 protein [36]. Fahlgren et al. (2013) proposed that the 25-nucleotide small RNAs produced by Phytophthora were involved in silencing repetitive genetic elements, and miRNAs might regulate the expression of amino acids/auxin permeases [37]. However, the identification and functional verification of endogenous ncRNAs have rarely been reported in notorious *P. expansum*. The further exploration of the potential regulatory networks involved should be encouraged as well.

In the present study, the global expression profiles of mRNAs, lncRNAs, miRNAs, and circRNAs in *P. expansum* spores and mycelia were determined using RNA-sequencing technology. The potential functions, the involved pathways, and the interaction networks were predicted by bioinformatics analyses as well. The results will lead to a better understanding of the regulatory machinery of the growth of *P. expansum* and provide a theoretical basis to develop novel control strategies at the molecular level.

## 2. Materials and Methods

### 2.1. Pathogen and Fungal Phenotype

*Penicillium expansum* (CGMCC3.3703) was activated by inoculating on an apple fruit (Malus domestica Borkh cv. Red Fuji). Then, the pathogen was reisolated and maintained on potato dextrose agar (PDA) medium at 25 °C. Images of the fungal colony and the apple fruit with a typical blue mold symptom were captured by an EOS 60D digital camera (Canon, Tokyo, Japan).

A suitable aliquot of fresh spore suspension was added to potato dextrose broth (PDB) medium with a final concentration of 1.0 × 10^6^ spores L^−1^ in a 100 mL system. After 6 or 18 h of culturing at 25 °C under a 200 rpm shaking condition, fungal spores and mycelia were collected by centrifugation. Then, 4′, 6′-diamidino-2-phenylindole (DAPI) with final concentrations of 50 mg L^−1^ were used to stain samples, as described by Lai et al. (2017) [38]. The Eclipse Ni-U microscope (Nikon, Tokyo, Japan) with individual filter sets was used to capture the images.

### 2.2. Sample Preparation and Quantitative Validation

The pathogen was routinely cultured on PDA plates for two weeks at 25 °C. Conidial suspension was obtained by flooding the sporulating cultures of *P. expansum* with sterile distilled water containing 0.05% (*v*/*v*) Tween-20. To remove any adhering mycelia, spore suspensions were filtered through four layers of sterile cheesecloth. Then, a suitable liquot of fresh spore suspension was added to a 250 mL conical flask containing 100 mL PDB to obtain a final concentration of 1.0 × 10^6^ spores/mL with the aid of a hemocytometer. After 6 or 18 h of culturing at 25 °C under a 200 rpm shaking condition, fungal spores and mycelia were collected by centrifugation, washed twice with sterile distilled water, and quickly frozen in liquid nitrogen. A sample with 6 h of culturing and a biological repeat were named Pe6h-1 and Pe6h-2, respectively. A sample with 18 h of culturing and a biological repeat were named Pe18h-1 and Pe18h-2, respectively. Subsequently, all samples were sent to Beijing Genomics Institute (BGI) Co., Ltd. (Beijing, China), for transcriptome sequencing. Meanwhile, samples were prepared in the same way for quantitative real-time PCR (qRT-PCR) validation. TRIzol Reagent (Invitrogen, Carlsbad, CA, USA), the FastQuant RT Kit (Tiangen Biotech, Beijing, China), and a 2× Ultro SYBr mixture (CW Bio, Taizhou, China) were applied to extract total RNAs, synthesize first-strand cDNA, and determine relative expression levels of target genes in order. Specific primer pairs for randomly selected genes (RSGs) from RNA-sequencing results are shown in Appendix A. Expression changes of RSGs were normalized by *β*-*tubulin* and evaluated using the 2^(−△△Ct)^ analysis method. The experiment was repeated twice.

### 2.3. LncRNA-Sequencing Analysis

LncRNA sequencing strictly followed the experimental operation protocol of the DNBSEQ platform. Briefly, total RNAs were firstly purified by removing rRNAs before fragmentation. Then, first-strand cDNA was generated using random hexamer primed reverse transcription, followed by a second-strand cDNA synthesis with dUTP instead of dTTP. The sequence ends were added, as was a ligated adaptor. Subsequently, the U-labeled second-strand template was digested by the UDG enzyme and PCR amplification was performed. After cyclization and nanoball synthesis, sequencing was conducted on the DNBSEQ platform.

For bioinformatics analysis, reads with low quality, adapter contamination, and a highly unknown N base were filtered by SOAPnuke (v1.5.2). Remaining clean reads were stored in FASTQ format, mapped to the reference genome (https://www.ncbi.nlm.nih.gov/genome/11336?genome_assembly_id=212204 (accessed on 11 October 2022)) by HISAT2 (v2.0.4), and assembled with StringTie (v1.0.4) and Cufflinks (v2.2.1) in order [39]. To distinguish between mRNA and lncRNA, three predictive software programs (eCPC (v0.9-r2), txCdsPredict (http://hgdownload.soe.ucsc.edu/admin/jksrc.zip), and CNCI (https://github.com/www-bioinfo-org/CNCI)) and the Pfam database (http://pfam.xfam.org/) (accessed on 15 October 2022) were used to predict the coding ability of the new transcripts (Appendix A). Bowtie2 (v2.2.5) was used to align clean reads to the reference sequence and count the coverage of genes. Then, mRNAs were annotated by the Nucleotide Sequence Database (NT), the Non-Redundant Protein Sequence Database (NR), the Clusters of Orthologous Genes Database (COG), the Kyoto Encyclopedia of Genes and Genomes database (KEGG), and the Swiss-Prot database. The Blast2GO (v2.5.0) and the NR annotation results were used for the Gene Ontology (GO) annotation, and an InterProScan5 (v5.11–51.0) was used for InterPro annotation (Appendix A).

RSEM (v1.2.12) was used to calculate the expression of transcripts [40]. Fragments per kilobase of the exon model per million mapped fragments (FPKM) represented the expression level of the gene (Appendix A). DEGseq (https://bioconductor.org/biocLite.R (accessed on 20 October 2022)) was used for differential analysis, and filtration conditions of genes with significant differences were as follows: fold change ≥ 2.00 and false discovery rate (FDR) ≤ 0.001 (Appendix A). Enrichment analysis of DE genes was performed based on the KEGG and Gene Ontology Consortium databases.

INFERNAL (v1.0) was used to annotate the lncRNA family based on the Rfam database (http://pfam.xfam.org/ (accessed on 20 October 2022)) according to the common ancestor of lncRNA at the evolutionary level (Appendix A) [41]. For lncRNA target gene prediction, a Spearman correlation coefficient ≥ 0.6 and a Pearson correlation coefficient ≥ 0.6 of lncRNA and mRNA were required. For *cis*- and *trans*-regulation analysis, it was determined to be *cis*-regulated mode if lncRNA was located within 20 kb upstream or downstream of the mRNA. On the other hand it was judged as a *trans*-regulated mode if the binding energy was <−30 of lncRNA and mRNA in RNAplex (v0.2) (Appendix A) [42].

### 2.4. Small RNA-Sequencing Analysis

Small RNA sequencing was performed by BGISEQ-500 technology of BGI following the standard protocol. Briefly, the 18–30 nt RNA segment was separated by polyacrylamide gel electrophoresis (PAGE). A 5′-adenylated, 3′-blocked single-stranded DNA adapter was linked at the 3′ end of selected small RNAs. After reverse transcription primer hybridization and ligation of the 5′ adaptor, the first-strand cDNA was synthesized. Then, PCR amplification and PAGE gel separation were conducted to purify and enrich cDNA of 100–120 bp. Finally, quantitation and pooling cyclization of the library were conducted.

After strict quality control, the remaining clean tags were stored in FASTQ format. Bowtie2 and Cmsearch were used to map clean reads to the reference genome or other sRNA (Appendix A). The miRDeep2 (v 2.0.1.2) and miRA (https://github.com/mhuttner/miRA (accessed on 10 November 2022)) software was used to predict novel miRNA by exploring the characteristic hairpin structure of the miRNA precursor (Appendix A) [43,44]. The siRNA prediction followed the criteria that a double-stranded RNA was 22–24 nt in length, each strand of which was 2 nt longer than the other (Appendix A) [45]. The small RNA expression level was represented by transcripts per million (TPM). DEGseq was used to determine DE genes when the fold change was ≥2.00 and the Q-value was ≤0.001 (Appendix A). TAPIR (http://bioinformatics.psb.ugent.be/webtools/tapir/), psRobot (v1.2), and TargetFinder (https://github.com/carringtonlab/TargetFinder) (accessed on 10 November 2022) were used to predict miRNA target genes (Appendix A) [46,47]. GO and KEGG Enrichment analyses of miRNA target genes were performed when a corrected *p*-value ≤ 0.05 was taken as a threshold. The Pearson correlation coefficient of miRNAs and target mRNAs was calculated, and the networks were established by VisNetwork. Furthermore, Pearson’s correlation test and a hypergeometric distribution model were used to calculate the correlation between DE miRNAs, DE lncRNAs, and DE mRNAs. The results are shown in Appendix A.

### 2.5. CircRNA-Sequencing Analysis

The circRNA library preparation followed the experimental procedure of the DNBSEQ platform. Briefly, RNA samples were purified by removing rRNAs and degrading linear RNAs. The remaining circular RNAs were fragmented and cDNA synthesis was performed. Then, the adaptor was ligated to the 3′ ends. After PCR amplification and denaturation, the circularization of single-stranded PCR products was set up. Rolling cycle amplification was used to generate, load into patterned nanoarrays, and sequence the DNA nanoballs through combinatorial probe–anchor synthesis.

After quality control, CIRI (v2.0.5) and find_circ (v1.2) were used to predict circRNA [48,49]. The known circRNAs were identified according to the circBase database (http://www.circbase.org (accessed on 10 November 2022)), and the functional annotations were based on source genes of the circRNAs (Appendix A). The expression level of circRNA was calculated using reads spanning the junction site with at least 10 bp coverage at both ends. Differential expression analysis was performed using a DEGseq (Q-value ≤ 0.001 or an absolute value of Log_2_FoldChange ≥ 1) (Appendix A). GO and KEGG enrichment analyses of source genes of the differentially expressed circRNAs were performed when a corrected *p*-value ≤ 0.05 was taken as a threshold.

### 2.6. Statistical Analysis

Statistical analysis was conducted with Microsoft Excel 2016 (v16.0.4266.1001) software. Analysis of variance was used for comparing more than two means. Differences at *p* < 0.05 were considered to be significant.

## 3. Results

### 3.1. The Phenotype of P. expansum at Different Developmental Stages

After 6 h of culturing in PDB medium, spores of *P. expansum* were rounded to ellipsoidal, turned greenish, were smooth-walled and became larger in volume. Meanwhile, one or two nuclei were clearly visible in the spores (Figure 1A). This indicates that the spores had been activated from dormancy; however, the cellular differentiation and polar growth had not started. After 18 h of culturing, the spores had finished germination, germ tube elongation, and hyphae formation. The genetic materials had also been replicated many times and multiple nuclei were distributed throughout the hyphae (Figure 1A). Spores and hyphae represented two typical developmental statuses and had remarkably different morphological characteristics. The differences end up determining the viability, resistance, and pathogenicity of *P. expansum*.

On PDA medium, colonies of *P. expansum* with thick white margins expanded quickly and presented a fasciculate texture. The surfaces were covered with massive light green spores (Figure 1B). After re-inoculation in apple fruit, the blue mold rot induced by *P. expansum* generated brownish and slightly sunken lesions. White mycelial growth across the surface could be observed at first, with the accumulation of abundant conidia following (Figure 1C). Therefore, the pathogen used in this experiment presented a typical phenotype of *P. expansum* and showed strong pathogenicity and vigorous growth.

### 3.2. LncRNA-Sequencing Results

Through the DNBSEQ platform, four *P. expansum* samples were analyzed and each sample generated about 11.50 Gb of data. A total of 16,164 transcripts were detected, and the genomic mapping rate was more than 71.47% (Appendix A). Among them, 2585 novel mRNAs and 3362 novel lncRNAs were found after coding-capacity prediction (Appendix A). The novel mRNA annotation results are shown in Appendix A. Most of them possessed catalytic or binding activity and became involved in the metabolic process and the cellular process. The number of gene products belonging to membrane components was the highest.

The expression levels of mRNAs and lncRNAs are shown in Figure 2 and Appendix A. Compared with the results from qRT-PCR and the transcriptomic sequencing of nine randomly selected genes (RSGs), the change trends between Pe6h and Pe18h were similar in the two detection methods (Figure 3). Two different methods of regression analysis of RSG expression changes were also performed, and the Pearson correlation coefficient was 0.951 (Appendix A). This indicates that data obtained from the two approaches showed a good correlation and that the quantitative results from the transcriptomic sequencing were meaningful and valuable. Further, 1482 novel mRNAs (763 up-regulated and 719 down-regulated), 5987 known mRNAs (2561 up-regulated and 3426 down-regulated), and 2047 lncRNAs (1320 up-regulated and 727 down-regulated) were significantly differently expressed (DE) between Pe6h and Pe18h (Figure 2 and Appendix A). GO and KEGG enrichment analyses of DE mRNAs are shown in Figure 4. The relatively large numbers of genes were related to RNA transport and spliceosome, ribosome, and ribosome biogenesis. The genes involved in the cytosolic DNA-sensing pathway showed the highest rich factor value.

By comparing lncRNAs to miRbase, no potential miRNA precursors were detected in lncRNAs. The lncRNA family prediction was performed at the evolutionary level as well. The number of lncRNAs belonging to the RF01306, RF00223, and RF01295 families was relatively larger (Appendix A). The function of lncRNA was usually related to the nearest protein-coding gene. The adjacent mRNAs of lncRNAs could act as potential target genes due to the *cis* action mode. A total of 14,771 lncRNA target genes were predicted (Appendix A), and the overlap classification of lncRNAs and their target genes are shown in Figure 5. The number of classes of lncRNA anti-overlap mRNA and anti-complete mRNA in the exon was the highest. The expression levels of 4486 or 5147 DE lncRNA target genes were up-regulated or down-regulated between Pe6h and Pe18h (Appendix A). Through enrichment analysis of DE lncRNA target genes, it was determined that the GO enrichment results were similar to those of DE mRNAs. However, the KEGG enrichment results showed a certain difference from those of DE mRNAs. The genes involved in the notch-signaling pathway presented a higher rich factor value (Figure 6).

### 3.3. Small RNA-Sequencing Results

Using BGI-500 technology, more than 32,415,610 clean tags were acquired for each sample, and at least 48% of clean tags could be mapped to the reference genome (Appendix A). Finally, a total of 46 known miRNAs, 10 novel miRNAs, and 86 novel siRNAs were detected in four samples (Appendix A). The base distribution of known miRNAs, novel miRNAs, and novel siRNAs had no obvious regularity. For the first base distribution, the ratio of U was the highest and the ratio of G was the lowest (Appendix A). The stem-loop structure of novel miRNA precursors in *P. expansum* are shown in Appendix A. The expression levels of miRNAs and novel siRNAs are presented in Appendix A. A total of 40 miRNAs and 38 novel siRNAs were differentially expressed between Pe6h and Pe18h.

Using TargetFinder and psRobot, 1119 miRNA target genes were predicted (Appendix A). Based on expression levels in Pe6h and Pe18h, the correlation of 37 co-differentially expressed miRNA/target pairs were determined. Among them, 20 groups were positive correlations and others were negative. The regulatory networks are shown in Figure 7. Due to the increasing filtrating threshold, simplified networks are presented in Appendix A as well. As shown in Figure 7, novel_mir4, novel_mir7, and miR168a may have been involved in multiple pathways by interacting with different targets. The GO and KEGG enrichment analysis of co-differentially expressed mRNAs are shown in Appendix A. The results indicate the miRNA/target pairs played more prominent roles in the pathways of amino acid metabolism and protein transport.

In addition, it is well known that both lncRNA and mRNA are miRNA response elements. Therefore, miRNA target sites and binding energy with mRNA or lncRNA were firstly predicted. Then, Pearson correlation coefficients of miRNA/mRNA, miRNA/lncRNA, and lncRNA/mRNA were calculated to determine the significance of miRNA shared between lncRNA and mRNA (Appendix A). Finally, the cross-regulated networks of lncRNA–miRNA–mRNA were constructed and are shown in Figure 8. It is worth noting that miR168-5p and miR168b played essential roles in the development of *P. expansum* due to their central place in the networks.

### 3.4. CircRNA-Sequencing Results

CircRNAs sequencing was also performed using BGI-500 technology. More than 1.1 × 10^8^ Mb clean reads of each sample were acquired. After prediction using CIRI (v2.0.5) and find_circ (v1.2) software, 5349, 4592, 605, and 689 circRNAs were found in Pe6h-1, Pe6h-2, Pe18h-1, and Pe12h-2 (Appendix A). Most of them located in exon or intergenic regions (Appendix A). The expression level of all circRNAs is shown in Figure 9A, and a total of 9235 circRNAs (950 up-regulated and 8285 down-regulated) were differentially expressed between Pe6h and Pe18h (Figure 9B). The source genes of DE circRNAs were classified by GO and KEGG enrichment analysis (Figure 10). The results indicate that the number of DEGs involved in ribosomes, biosynthesis of amino acids, endocytosis, carbon metabolism, and RNA transport were larger. Meanwhile, DEGs involved in o-glycan biosynthesis, ribosomes, and oxidative phosphorylation presented high rich factor values. For pathway enrichment analysis, a higher percentage of DEGs belonged to the metabolism term, especially to carbohydrate, amino acid, lipid, and energy metabolism. In addition, a certain number of DEGs were involved in translation, transport, and catabolism.

## 4. Discussion

*P. expansum* is the most popular and economically significant post-harvest pathogen. Due to the detrimental effects on the pome fruit industry and human health worldwide annually, more efficient and eco-friendly methods and strategies against this necrotrophic fungus are urgently needed. For this purpose, many fundamental studies regarding the development and virulence of *P. expansum* have been conducted at the biochemical, physiological, and molecular levels. With the advent and continuous upgrading of next-generation sequencing technology, genomic resources of *P. expansum* with high-quality assembly have been obtained, which contributes to the identification and functional analysis of the key genes or proteins involved [50,51].

The spores of *P. expansum* were highly specialized cells that were easily separated from parent fungi and disseminated for great distances via multiple media. Once having contacted suitable hosts, subsequent germination and a new somatic phase occurred. As either the candidates for initiating a new generation or the final products of a reproductive process, they were unique in the fungal life cycle. The mycelia of *P. expansum* were the vegetative parts consisting of a mass of branching filamentous hyphae. They were developed from the germinating spores, had specialized structures, and played important roles in absorbing nutrients and infecting hosts. Based on this, a comparative transcriptomic data set between spores and mycelia of *P. expansum* was generated in the present study. A total of 2595 novel mRNAs, 3362 lncRNAs, 10 novel miRNAs, 86 novel siRNAs, and 11,238 circRNAs were predicted and quantified. Of these, 1482 novel mRNAs, 5987 known mRNAs, 2047 lncRNAs, 40 miRNAs, 38 novel siRNAs, and 9235 circRNAs were differentially expressed between spores and mycelia. The annotation of transcripts, lncRNA family classification, lncRNA and miRNA target gene prediction, stem–loop structure prediction of novel miRNAs, and circRNA source gene annotation were conducted subsequently. The GO and KEGG enrichment statistics of DE mRNAs, DE lncRNA target genes, DE miRNA target genes, and DE circRNA source genes were performed as well. In addition, based on the quantitative analysis and the Pearson coefficient calculation, the interaction between DE miRNA and DE mRNA, and possible cross-regulated networks of lncRNA–miRNA–mRNA in *P. expansum* were predicted.

Many transcriptomic data sets of *P. expansum* were available for exploration of the *P. expansum* pathosystem, and many genes involved in the infection process, germination, and different stress responses have been described [11,52]. In this study, compared with those in spores, the expression levels of 7464 mRNAs (containing 1482 novel mRNAs) were significantly changed in mycelia. Most of the DEGs participated in the major metabolic pathways (such as protein metabolism, nucleotide metabolism, and carbohydrate metabolism) and led to morphological changes in *P. expansum*. Further interpretation of the data led to us noticing many valuable issues. For example, DEGs involved in the cytosolic DNA-sensing pathway presented the highest rich factor values. The members belonging to this pathway were specific families of pattern recognition receptors. They responded to cytosolic DNA arising from invading microbes or host cells and generated innate immune host defense mediated by type I interferon and cytokines. It is suggested that, following the growth of the pathogen, mycelia have prepared to face biotic and abiotic stresses such as viral infections and host innate immunity. This is just one case of many questions raised from the enrichment analysis of DEGs between spores and mycelia. After comprehensively integrating and jointly investigating at multiple levels, it was determined that the transcriptomic results would prompt more thoughts and inform valuable discussions.

LncRNAs are transcribed RNA molecules longer than 200 bases and not translated into proteins. Significant progress has been made toward the understanding of the mechanistic and functional roles of lncRNAs across species and various cell types. They play crucial roles in transcription and translation regulation, RNA metabolism, epigenetic control and signaling through *cis* interactions with RNAs, and through *trans* interactions with DNA, RNAs, or proteins [53,54,55]. Based on the accumulation of genomic knowledge and the expanding lncRNA repository, the identification and functional verification of lncRNAs in phytopathogens were also conducted [56,57]. Donaldson and Saville (2013) identified 204 natural antisense transcripts in Ustilago maydis, of which as-um02151 could positively influence pathogenesis in both seedling and cob infections of Zea [58]. Aarthanari et al. (2014) identified 939 novel lncRNAs with 477 antisense transcripts in *Neurospora crassa* and demonstrated the ability of lncRNAs to respond to environmental stimuli [59]. Yang et al. (2019) found that lncRNA *AOANCR* could affect citrinin production by negatively regulating *mraox* gene expression in *Monascus purpureus* M9 [60]. However, no reports referred to the identification and biological function of lncRNAs in *P. expansum*. Therefore, 3362 predicted lncRNAs in total with high confidence through strict filtration in this study would provide new possibilities and opportunities to thoroughly analyze the pathosystem of *P. expansum* in RNA levels. In particular, the properties, expression profiles, and involved pathways of DE lncRNAs would facilitate further exploration of their biological functions and interaction network with other molecules in the development of *P. expansum*.

RNA interference is a highly conserved regulatory system that functions to inhibit the expression of specific genes in eukaryotes. Generally, this process involves miRNAs and siRNAs, both of which are small ncRNAs produced by endoribonucleases and that guide the RNA-induced silencing complex (RISC) to the target, yet they possess a distinct action mode [61,62]. miRNAs are usually transcribed by RNA polymerase II in the nucleus, and pre-miRNAs contain a stem-loop structure. In the cytoplasm, pre-miRNA is processed into an miRNA duplex by Dicer (RNase type III) and incorporated into RISC with the Ago protein. The matured miRNA (20–24 nucleotides) can recognize the 3′ untranslated region (UT) of mRNA with a complementary sequence and silence the target genes by cleavage, deadenylation, or translational inhibition. The single miRNA can have multiple targets and the same mRNA can be regulated by different miRNAs [63]. On the other hand, siRNAs are a kind of double-stranded RNAs with 20–25 nucleotides. They are firstly generated by RNA-dependent RNA polymerase (RdRP) and cleaved by Dicer, resulting in the specific 3′ end of each strand with two unpaired nucleotides. A selected guide strand combines with the Ago family to form RISC and lead RISC to homologous mRNA. Only one mRNA could be specifically targeted, and then the gene was silenced by RNA degradation, translational repression, and transcriptional silencing induced by transcriptional silencing initiated by the changing chromatin state [30,64]. Many reports have proven that miRNAs and siRNAs play crucial roles in cell differentiation, chromosome structure modification, and epigenetic regulation in filamentous fungi. For example, Lee et al. (2010) demonstrated that miRNA-like small RNAs (milRNAs) and Dicer-independent siRNAs were generated by diverse pathways in *Neurospora crassa* [65]. Lin et al. (2015) identified 4 conserved miRNAs and 63 novel milRNAs in *Antrodia cinnamomea*, several of which were associated with tri-terpenoid synthesis, mating-type recognition, chemical or physical sensing, and transportation [66]. Wang et al. (2017) found that microRNA-like RNA 1 was a crucial pathogenicity factor of *Puccinia striiformis* f. sp. *tritici* that could decrease the resistance of wheat by suppressing the pathogenesis-related 2 gene [67]. Jin et al. (2019) showed that *VdHy1* is essential to the virulence of *Verticillium dahliae*, and an milRNA VdMILR1 could inhibit its expression by increasing the histone H3K9 methylation of *VdHy1* [68]. Hu et al. (2022) found that Ago1-associated miRNA miR8788 of *Phytophthora infestans* could target *StABH1* (an alpha/beta hydrolase-type encoding gene of potato) and promote potato late blight disease [69]. For siRNAs, the findings of Khatri and Rajam (2007) showed that siRNA could induce a specific silencing effect, leading to the decline of target mRNA titers and cellular polyamine concentrations and mycelial growth reduction [70]. Hammond et al. (2013) uncovered that SAD-4 and SAD-5 proteins in *Neurospora crassa* were important for masiRNA production, which is associated with meiotic silencing [71]. To make up for a deficiency in related research on *P. expansum*, conserved and novel miRNAs and siRNAs were identified in different growth stages. Through expression profiles and target prediction of miRNAs, the relationships between miRNAs and fungal morphological-change miRNAs have been preliminarily understood. After further analysis based on expression levels, the correlation between DE miRNAs and DE mRNAs in regulatory networks has also been determined. Among them, miR168a-5p, miR396g-5p, novel-mir4, and novel-mir7 presented more potential to be proceeded with due to the more complex networks involved.

It is worth noticing that lncRNAs can participate in transcription regulation, epigenetic effects, and other complex mechanisms to modulate various physiological and pathological processes [28]. Meanwhile, lncRNAs can competitively adsorb miRNAs like a sponge, which weakens the functions of miRNAs at the post-transcriptional level [72]. Therefore, an lncRNA–miRNA–mRNA regulatory network was established using the acquired expression profiles of lncRNAs, miRNAs, and mRNAs in spores and mycelia. The results contribute to some new insights explaining the complex regulatory network at RNA levels. The most obvious examples are miR168 family member-centered networks, which were closely related to the development of the pathogen and contained a large amount of interaction information among lncRNA, miRNA, and mRNA. The miR168 family with two orthologs (miR168a and miR168b) and structural flexibility was highly conserved in plants, and a total of 49 sequences belonging to 31 species had been included in the miRBase (Version 22.1) [73,74,75]. miR168 functioned on the entire miRNA pathway by directly regulating the central miRNA effector AGO1, the catalytic subunit of RISC. A complex feedback loop containing miR168-guided cleavage of AGO1 transcripts and AGO1-mediated stabilization of miR168 indicated that the homeostasis of both of them was essential for the miRNA-mediated regulation [76,77]. As a pivotal regulator, miR168 could contribute to leaf epinasty, anthesis, fruit development, yield, response to nutritional deficiency, and immunity to fungal or viral elicitors [78,79,80,81,82]. Moreover, exogenous plant miR168a can regulate target gene expression in mammals. MIR168a in rice can suppress the expression of the low-density lipoprotein receptor adapter protein 1 gene in liver and influence the low-density lipoprotein removal in human and mouse [83]. Akao et al. (2022) also reported that hvu-MIR168-3p increased the RNA and protein expression levels of glucose transporter 1 by inhibiting the expression of the genes associated with mitochondrial electron transport chain complex I in human cells [84]. These interesting results indirectly prove the significant roles of miR168 in fungal development and encourage us to pay attention to the homeostasis of AGO1 and other miRNA levels when disturbing the proper functioning of the miR168 pathway using mutations or transgene approaches. Therefore, the association between fungal development and the lncRNA–miRNA–mRNA regulatory network derived from transcriptomic results is worthy of further exploration and discovery.

In addition, as novel regulatory RNAs, circRNAs possess covalently closed loop structures without either 5 to 3 polarity or a polyadenylated tail. They are generated through alternative splicing of exons depending on the existence of RISC and take part in diverse physiological and pathological processes in various cell lines and across different species. Growing evidence shows that circRNAs can act as sponges for miRNAs, regulate the alternative splicing and expression of source genes, induce the formation of pseudogenes, and have potential translational ability [85]. However, there are relatively few reports on the identification, action mechanism, and functional characterization of circRNAs in plant fungi. Therefore, based on next-generation sequencing data, 11,238 circRNAs were predicted in *P. expansum*, and the expression profiles of circRNAs in spores and mycelia were also determined. Through the GO and KEGG enrichment analyses of source genes of DE cirRNAs, the tissue specificity, stability, versatility, and high expression levels of cirRNAs were determined, and the possible relationship between circRNAs and fungal growth was preliminarily exhibited as well. These results may expose new perspectives for the detailed molecular dissection of the development of *P. expansum* and provide the basis for integrating regulatory networks with other types of RNA molecules and proteins.

In conclusion, a comparative transcriptomic analysis between spores and mycelia of *P. expansum* was performed. The identified novel non-coding RNAs, corresponding expression profiles, biological processes involved, and predicted interaction networks among them will contribute to exploring the regulation mechanisms on the development of *P. expansum* on RNA levels.

## Figures and Tables

**Figure 1 jof-09-00999-f001:**
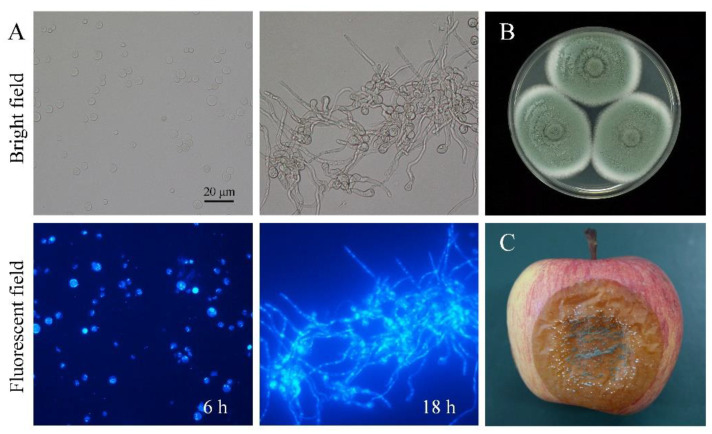
The colonial mycelial (**A**) and colonial (**B**) morphological features of *P. expansum* and symptoms of blue mold on apple (**C**). (**A**) The spores and hyphae stained by 4′, 6′-diamidino-2-phenylindole (DAPI) after 6 h and 18 h of culturing in PDB medium; (**B**) the colonies of *P. expansum* on PDA medium; (**C**) the symptom of blue mold on apple.

**Figure 2 jof-09-00999-f002:**
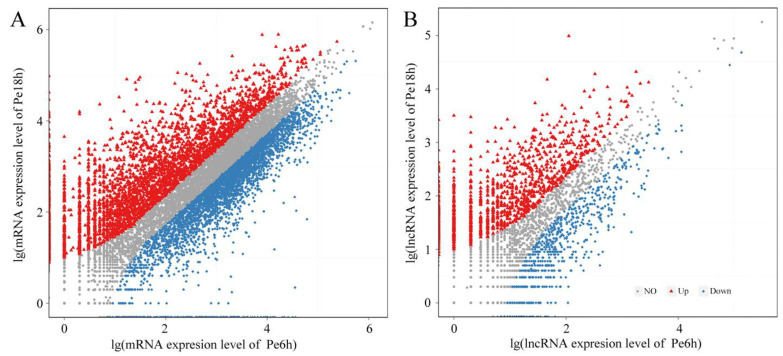
The scatter plot of the differentially expressed (DE) mRNAs (**A**) and lncRNAs (**B**) between Pe6h and Pe18h. Blue squares represent the down-regulated genes, red triangles represent the up-regulated genes, and grey dots represent the not significantly different genes.

**Figure 3 jof-09-00999-f003:**
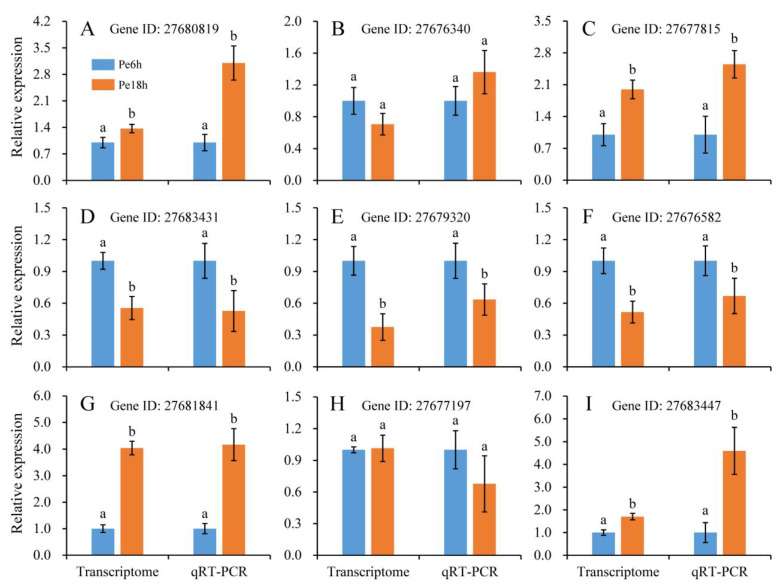
The relative expression levels of the randomly selected genes (RSGs) in Pe6h and Pe18h resulted from transcriptomic sequencing and qRT-PCR approaches. (**A**–**I**) show the relative expression levels of *RSG1* to *RSG9*, respectively. The detailed information on *RSG1* to *RSG9* is shown in Appendix A. Bars indicate the standard deviation of the means. Lowercase letters a and b indicate significant differences at *p* < 0.05 based on Student’s *t*-test between Pe6h and Pe18h samples, respectively, for different detection methods.

**Figure 4 jof-09-00999-f004:**
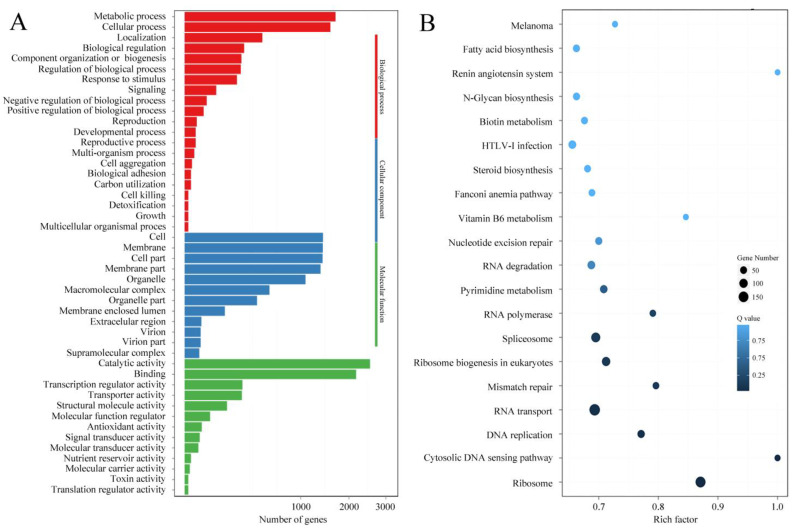
GO and KEGG enrichment analyses of differentially expressed genes (DEGs) between Pe6h and Pe18h. (**A**) GO enrichment results. Different colors represent different GO classifications. (**B**) KEGG enrichment results. The deepest color indicates the highest confidence. The rich factor is the ratio of the number of differentially expressed genes and the total identified genes in the same KEGG. The dot size indicates the number of DEGs.

**Figure 5 jof-09-00999-f005:**
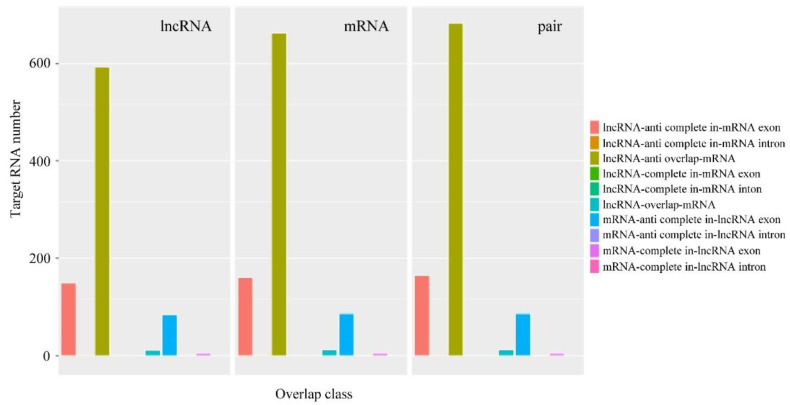
Overlap classification of lncRNAs and mRNAs involved in *cis*-regulation in *P. expansum*. The function of lncRNA is related to the protein-coding gene near cis so it can serve as a candidate target gene, thus screening the adjacent mRNA of lncRNA as a target gene. The regulation is not a one-to-one correspondence, and one lncRNA regulates one mRNA called a pair. A detailed classification of overlap on lncRNAs and mRNAs is shown. The *X* axis indicates the different cis classification with different colors, and the *Y* axis indicates the number of target RNAs.

**Figure 6 jof-09-00999-f006:**
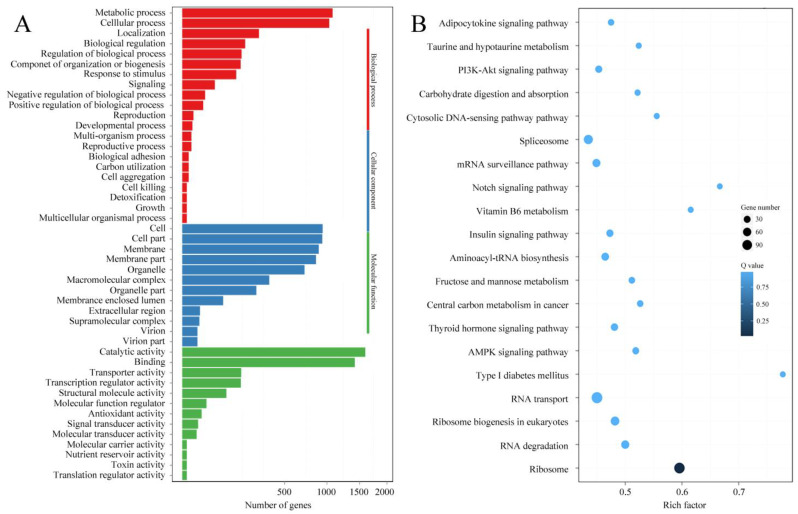
GO and KEGG enrichment analyses of differentially expressed lncRNA target genes between Pe6h and Pe18h. (**A**) GO enrichment results. Different colors represents different GO classifications. (**B**) KEGG enrichment results. The deepest color indicates the highest confidence. The rich factor is the ratio of the number of differentially expressed genes and the total identified genes in the same KEGG. The dot size indicates the number of DEGs.

**Figure 7 jof-09-00999-f007:**
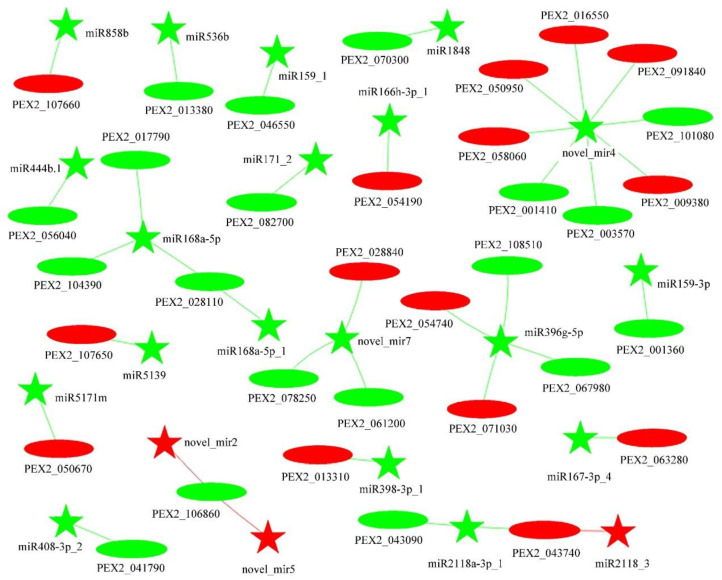
Interaction networks between DE miRNA and DE mRNA between Pe6h and P18h. The stars represent the DE miRNAs and the ovals represent the DE mRNAs. Red and green represents up-regulation and down-regulation, respectively between Pe6h and Pe18h.

**Figure 8 jof-09-00999-f008:**
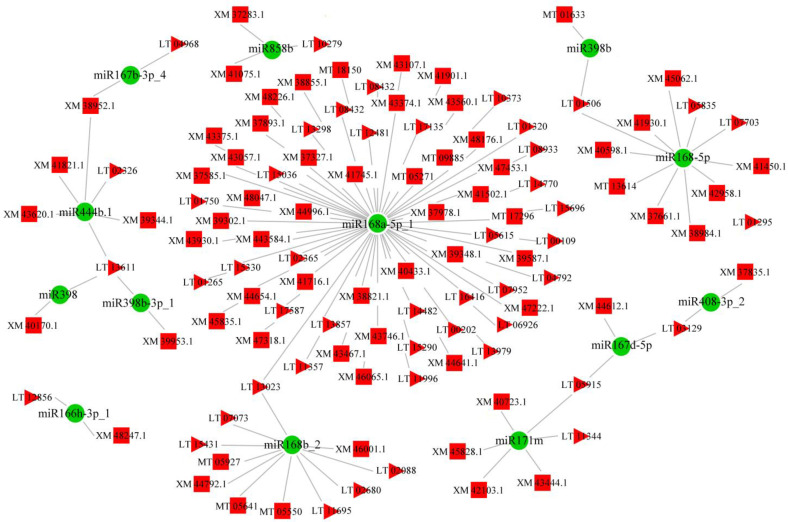
Interaction networks between DE miRNAs, DE mRNAs, and DE lncRNAs between Pe6h and P18h. The squares represent the DE lncRNAs, the triangles represent the DE mRNAs, and the green dots represents the DE miRNAs. XM, LT and MT stand for XM_0167, LTONS_000 and MTONS_000, respectively.

**Figure 9 jof-09-00999-f009:**
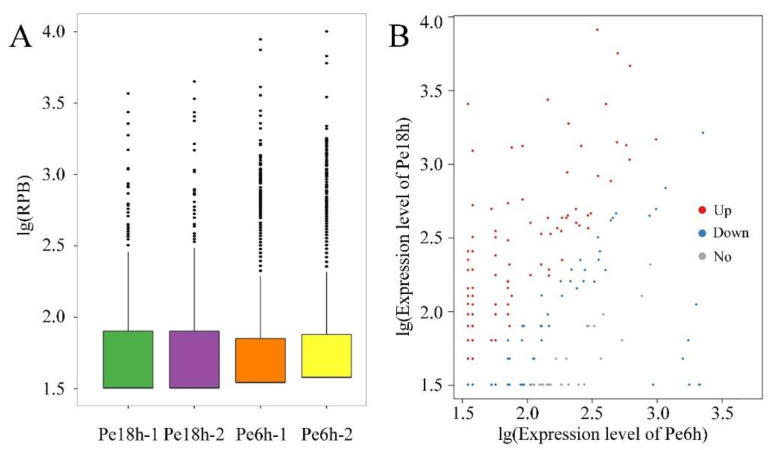
The expression analysis of circRNAs. (**A**) The box plot of circRNA expression levels in each sample. (**B**) The scatter plot of the differentially expressed circRNAs between Pe6h and Pe18h. Blue squares represent the down-regulated genes, red triangles represent the up-regulated genes, and grey dots represent the not significantly different genes.

**Figure 10 jof-09-00999-f010:**
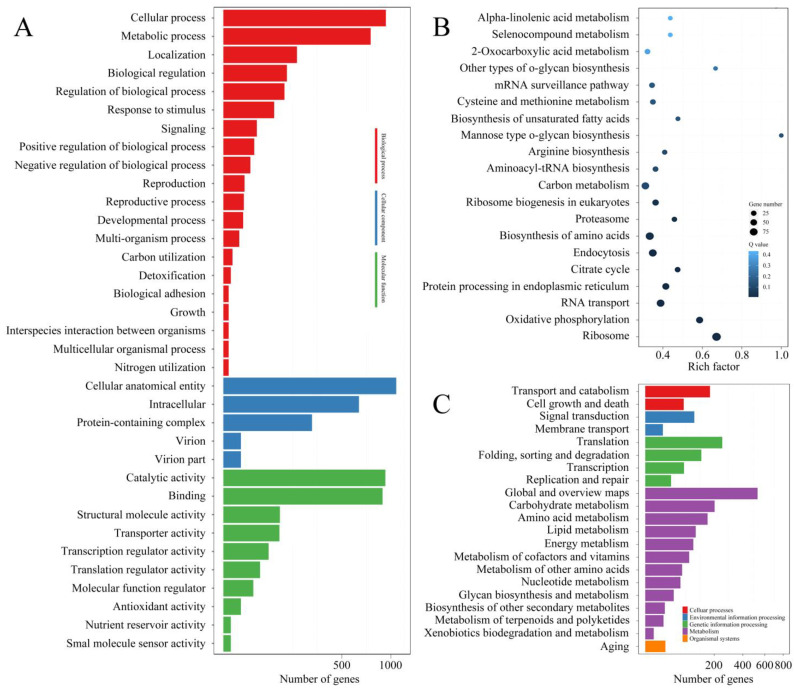
The GO and KEGG enrichment analyses of circRNA source genes. (**A**) GO enrichment results. Different colors represent different GO classifications. (**B**) KEGG enrichment results. The deepest color indicates the highest confidence. The rich factor is the ratio of the number of differentially expressed genes and the total identified genes in the same KEGG. The dot size indicates the number of DEGs. (**C**) Pathway enrichment results. Different colors represent different pathway classifications.

## Data Availability

The data presented in this study are available in article and Appendix A.

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
