# Peer review of "The Identification and Comparative Analysis of Non-Coding RNAs in Spores and Mycelia of Penicillium expansum"

_jof, 2023, doi:10.3390/jof9100999_

Round 1

Reviewer 1 Report

In general, the presented work is scientifically interesting. The identification of deferentially expressed genes in several fungal processes are important to determine pathological mechanisms such as fruit infection. However, this manuscript must be improved. 

Major Comments:

The identification of mRNA deferentially expressed between spores and mycelia of the fungus P. expansum, showed several mRNAs and lncRNAs changing their expression. At least 9 of these genes were randomly chosen and evaluated by qPCR to confirm transcriptomic data. However, transcriptomic of the other non-coding RNAs, such as miRNAs, are not validated through qPCR or Northern Blot. It is fundamental to demonstrate that bioinformatic data are coherent with experimental results. 

Minor Comments:

1. Figure 3 requires the addition of the name or code of each genes. I think that use RSG is not enough and it is hard to understand these results. It is importan identify in the same figure if the RSG is a mRNA or a lncRNA.

2. The legend of Figure 5 is very short. It is necessary the addition of relevant information to expand the legend. 

Reviewer 2 Report

Pathogenecity is a complex mechanism. In this study, the main sample  isolation of fungus from infected apple not clearly explain. How about environmanetal issue, media issue, how about normal fungal gene expression that not infected with apple did not well explain or ignored . Comments can find the the manuscript.

Round 2

Reviewer 1 Report

The new version of the manuscript is better than the original. Congratulations to the authors for the improvement of the work.

Reviewer 2 Report

Received the corrections